# Zero-Shot Data Maps.
# Efficient Dataset Cartography Without Model Training

**Angelo Basile**
angelo.basile@symanto.com
Universitat Politècnica de València
Symanto Research

**Marc Franco-Salvador**
marc.franco@symanto.com
Symanto Research

**Paolo Rosso**
prosso@dsic.upv.es
PRHLT Research Center
Universitat Politècnica de València

## Abstract

Data Maps (Swayamdipta et al., 2020) have emerged as a powerful tool for diagnosing large annotated datasets. Given a model fitted on a dataset, these maps show each data instance from the dataset in a 2-dimensional space defined by a) the model's confidence in the true class and b) the variability of this confidence. In previous work, confidence and variability are usually computed using training dynamics, which requires the fitting of a strong model to the dataset. In this paper, we introduce a novel approach: Zero-Shot Data Maps based on fast bi-encoder networks. For each data point, confidence on the true label and variability are computed over the members of an ensemble of zero-shot models constructed with different — but semantically equivalent — label descriptions, i.e., textual representations of each class in a given label space. We conduct a comparative analysis of maps compiled using traditional training dynamics and our proposed zero-shot models across various datasets. Our findings reveal that Zero-Shot Data Maps generally match those produced by the traditional method while delivering up to a 14x speedup. The code is available at https://github.com/symanto-research/zeroshot-cartography.

## 1 Introduction

Practices like annotation crowdsourcing and large-scale web scraping have enabled NLP researchers to create large annotated corpora (Snow et al., 2008; Brown et al., 2020). Concurrently, this growth in data volume has led to an increased prevalence of label noise and annotation artifacts within datasets (Abad and Moschitti, 2016; Gururangan et al., 2018). Further complicating matters, the widening gap between dataset annotators and end-users — researchers and practitioners who use the datasets to train models — often transforms these datasets into black boxes (Paullada et al., 2020). This division of labour, while economically advantageous in terms of time and resources, hinders thorough comprehension of the data. Dataset Cartography using training dynamics (Swayamdipta et al., 2020) offers a partial solution to this problem as it can help interpreting annotated corpora with respect to a model. Specifically, data maps reveal three loose groups of instances within an annotated dataset: easy-to-learn, ambiguous and hard-to-learn instances; the latter often correspond to annotation errors. Training new models based on samples from these regions can lead to better models with improved generalization. While cartography based on training dynamics can generally compile high quality maps, it does so at the cost of training a large model on the target dataset.

In this work, we introduce a novel, fast method for drawing Data Maps using zero-shot models. This approach capitalizes on the sensitivity of zero-shot models to label description framing, offering a more efficient and resource-friendly alternative to the original training dynamics method. Our approach allows for the generation of data maps without the requirement of extensive computational resources for the training of a robust model on a large dataset. When evaluated on data selection and error detection tasks, our experimental results show that zero-shot map coordinates are as useful as coordinates obtained through training dynamic maps, while requiring a fraction of the time to compute. Zero-shot data maps speed up the analysis of large datasets, removing the need for model training; they are a handy tool for industry practitioners managing daily unique datasets.

**Research Questions** In this study, we aim to answer the following research questions (RQ):

- **RQ1**: Can bi-encoder models serve as the ba-

sis for a lightweight, low-resource variant of data cartography? In particular, can map coordinates obtained through training dynamics be approximated by zero-shot models?

- **RQ2**: Can the confidence metrics from ensembled zero-shot models serve as a reliable indicator for finding annotation errors?

**Contributions**    We introduce a novel method for compiling data maps up to 14x faster. We compare our method against training dynamics-based maps. Furthermore, we show that zero-shot maps can be used for automatic annotation error detection. Finally, we will provide an easy-to-use implementation of our method for enabling NLP practitioners to inspect new datasets.

## 2    Data Maps with Zero-Shot Models

In this section, we first explain how our chosen zero-shot architecture works and why we chose it over other options (Section 2.1). We then proceed to illustrate the sensitivity of zero-shot models to label descriptions (Section 2.2). Finally, we describe how we can use this feature to build Data Maps with zero-shot models, thus avoiding model training (Section 2.3).

### 2.1    Zero-Shot Classification with Bi-Encoder Models

Recent work in NLP (Radford et al., 2019) have shown that large pre-trained language models (LLMs) can effectively be used in a zero-shot setup, meaning that they require no data to solve a task. Among the different architectures that have been shown capable of turning LLMs into zero-shot classifiers (Radford et al., 2019; Yin et al., 2019; Schick and Schütze, 2021), we focus on the bi-encoder (or Dual Encoder) class of models (Reimers and Gurevych, 2019; Mueller et al., 2022) due to their efficiency during inference. Bi-encoder models work as classifiers thanks to *label descriptions*, which we define as a natural language representation of each class in the label space. Label descriptions are essentially the sentence-embedding counterpart to the use of task descriptions with pre-trained language models (Radford et al., 2019). For example, consider a binary classification task to identify spam emails. Here, the label description for the spam class might be *This email contains unsolicited content* while for the non-spam class, it could be *This email includes relevant information.*

A bi-encoder model operates in a two-step process when given input texts and label descriptions: first, it generates embeddings for both the documents and label descriptions, and secondly, it uses these similarity scores as a classification function.

More formally, given a corpus of $N$ documents and a task with with $K$ classes — each with a corresponding label description $l_k$ — bi-encoders firstly independently embed the input texts and the label descriptions. We define the document embeddings as $X \in \mathbb{R}^{N \times d}$ and the label description embeddings as $L \in \mathbb{R}^{K \times d}$, where $d$ is the dimension of the bi-encoder embedding model $E(\cdot)$. Next, a similarity score is computed for each document and label description pair: this is done using a symmetric function $s(\cdot, \cdot)$ like cosine similarity or dot product. Finally, the class associated with the label yielding the highest similarity with the input text is chosen as the prediction:

$$\hat{y} = \text{argmax}_{l_i} s(E(x), E(l_i)) \qquad (1)$$

The application of a softmax transformation to the array of similarities, turns the bi-encoder model into probabilistic classifiers. These transformed similarities create a probability distribution over the labels:

$$P(y_i|x) = \text{softmax}(s(E(x), E(l_i))) \qquad (2)$$

In this way, the model provides not just a prediction but also a probability distribution over all labels, offering an interpretation of the model's confidence for each label.

We opt for the bi-encoder architecture because of its inference efficiency. In particular, bi-encoders can independently model input and labels, meaning that only a single pass over the input documents is necessary, regardless of the number of classes or label descriptions per class. In contrast, cross-encoder Natural Language Inference (NLI) models (Yin et al., 2019), a popular architecture for zero-shot classification, implies $K$ passes over the input data for each class $k$ within the label space. When defining multiple label descriptions $L$ per class to create a source of variation in predictions — as we do in this work — $N \times L \times K$ passes would be required, iterating for each class and each label description. In contrast, Mueller et al. (2022) demonstrate that bi-encoders can achieve quality similar to that of cross-encoder models while being

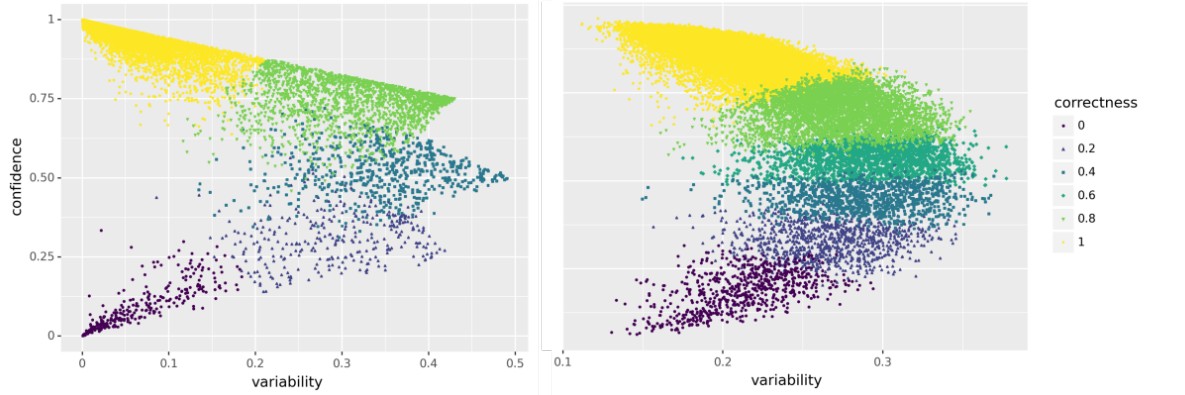

Figure 1: Data Maps for the training split (limited to 25k examples) of the *SST2* dataset, computed using training dynamics (*left*) and our zero-shot proposed method (*right*) with *sentence-t5-large* (Ni et al., 2022).

up to 20 times faster in terms of processed tokens per second.

## 2.2 Sensitivity to Label Description Phrasing

| | LABEL DESCRIPTION | F1 |
|---|---|---|
| a) | *Just bad!*
*All in all, the movie was awesome.* | 89.33 |
| b) | *Just terrible!*
*All in all, the film was great.* | 68.71 |
| | mean | 78.55 |
| | min | 24.79 |
| | max | 89.33 |

Table 1: Top: *a)* the best performing combination of label descriptions (red and green for the negative and positive class respectively) on the *SST* dataset and *b)* a semantically similar, but much worse, set of label descriptions. Bottom: Mean, minimum and max score across a set of 729 label description combinations for the *SST2* dataset.

LLM-based zero-shot models are notably sensitive to the phrasing and structure of the prompts used for specific tasks (Shin et al., 2020; Lu et al., 2022). Label descriptions suffer from the same problem. To illustrate this, let us consider the case of a sentiment analysis task with `positive` and `negative` as the class labels. The phrasing of label descriptions can vary within the same class. For instance, `positive` descriptions could be *All in all, the movie was awesome.* or *All in all, the film was great.*, while `negative` descriptions could be *Just bad!* or *Just terrible!*. Despite conveying the same meaning, the specific wording of these descriptions can have a substantial impact on the

model's performance, as shown in Table 1.

For each task, we first manually craft a large set of different label descriptions; next, we generate predictions for all the possible combinations of these label descriptions. More formally, let us assume we have a total of $L$ label descriptions per class $k$. For our input text we compute an embedding $x$. For each label description $i$ in class $k$, we calculate an embedding, denoted as $l_{c,i}$. Next, we apply a symmetric similarity function $f$ to determine the similarity between $x$ and each $l_{c,i}$, resulting in an array of similarities $S_k$ for each label description:

$$S_{k,i} = f(x, l_{k,i})$$

Hence, for each embedded input text $x$ and each class $k$, we generate an array of $L$ predictions (one for each label description). This leads to $N \times L \times C$ predictions for each input text, where $C$ is the number of classes and $L$ is the number of different label descriptions per class. In Figure 2 we show the distribution of macro F1 scores computed using a large set of different label descriptions[1] across several datasets: on average, performances range between state-of-the-art and purely random.

## 2.3 Zero-Shot Data Maps

Considering that bi-encoder classifiers a) deliver efficient inference time regardless of the number of labels, and b) exhibit performance variations based on the phrasing of label descriptions, we can translate the concept of training dynamics to zero-shot maps. More precisely, we compute confidence and variability across the predictions of $Z$ ensemble

---

[1]A detailed overview of the label descriptions can be found in Table 8 in the Appendix.

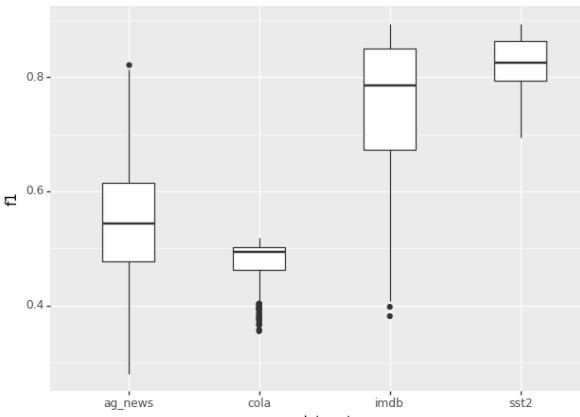

Figure 2: Zero-shot macro F1 performance of difference combinations of label descriptions for 4 datasets.

members of a zero-shot model instead of a series of training epochs. Intuitively, we can view zero-shot maps as the equivalent of a time-unrolled version of a training dynamics map: we take multiple predictions all at once from various zero-shot models, each prompted with a different label description, as opposed to gathering predictions from different epochs of the same model. Similar to the variable quality of predictions across training epochs, which generally improve as more epochs are completed, the performance of zero-shot models can also vary substantially depending on the label description, as we discuss in Section 2.2: this source of variation serves as the basis for computing confidence and variability.

The *average confidence* of a zero-shot model — defined by the embeddings of a specific set of label descriptions $\theta_z$ — over an instance $x_i$ with true label $y_i^*$ is defined as the average probability assigned by the ensemble to label $y_i^*$ across all the members $Z$ of the ensemble:

$$\mu_i = \frac{1}{Z} \sum_{z=1}^{Z} p_{\theta_{(z)}}(y_i^*|x_i)$$

Similarly, we define *variability* as the standard deviation of the model's probabilities across the ensemble members:

$$\sigma_i = \sqrt{\frac{\sum_{z=1}^{Z} (p_{\theta_{(z)}}(y_i^*|x_i) - \mu_i)^2}{Z}}$$

This measure quantifies the spread of model's probabilities across the ensemble, giving an indication of the degree of agreement or disagreement among the ensemble members about the correct label for an instance.

Finally, *correctness* simply tracks the number of times a model predicted the gold label. In Figure 1 we show an example of data maps compiled using both the training dynamics method and our novel zero-shot approach.

## 3 Experimental Setup

We first conduct a visual evaluation of the quality of our zero-shot data maps, comparing them with the maps obtained through training dynamics. Next, we follow the original data selection experiments and train a set of models on 8 sub-samples of three datasets. Finally, we evaluate zero-shot maps as error detectors and compare run time and resource usage against training dynamics maps.

### 3.1 Data Selection

For the data selection experiments, we first generate maps using both training dynamics and our proposed zero-shot method; for consistency, we use the same model — *sentence-t5-large* (Ni et al., 2022) — for both approaches. Next, following the existing literature (Swayamdipta et al., 2020), we define eight distinct regions based on all possible combinations of confidence, correctness and variability. These regions include:

- Easy-to-learn: These are instances that the models consistently predict with high confidence. These instances exhibit both high confidence and low variability.

- Hard-to-learn: These are instances that the models find challenging. These instances have low confidence and low variability.

- Ambiguous: These instances exhibit high variability and differing levels of confidence. They usually constitute the majority of the instances in a dataset.

- High confidence: Instances where the models, regardless of variability, exhibit consistently high confidence.

- High correctness: Instances that are consistently correctly predicted across different label descriptions or training epochs.

- Low correctness: Instances that are frequently misclassified.

- Low variability: Instances that yield consistent model predictions.

- Random: a baseline that randomly selects instances from all over the map

Given the map of the entire training set, we order the instances based on our strategy of interest (e.g., hard-to-learn, easy-to-learn, ambiguous, etc.), and then sample from this ordered list until we acquire 33% of the original dataset. In some cases, a class might not be represented in the sampled split: we avoid this by including four random instances from each class in the subset. Given a 33% sample of the original data, we train a classifier using exclusively such sample and evaluate its performance on the test split. For the zero-shot maps, we use the *sentence-t5-large* model as a frozen encoder and trained a logistic regression model on top.

## 3.2 Error Detection

A plausible interpretation for instances populating the low-confidence and low-variability map area might be to consider them as annotation errors. The interpretation is based on the idea that if a model persistently mispredicts the gold label as the most probable label for a given instance, it is likely that the gold label itself is incorrect. We test if the confidence coordinates provided by zero-shot data maps can serve as features for an automatic error detection system and compare with training dynamics data maps as a baseline. In their comprehensive study of automatic error detection, Klie et al. (2023) demonstrated that the confidence feature from training dynamics maps is as a robust error detection baseline, performing well across various tasks and datasets.

We test a simple linear model on the semi-automatically validated test subset of the IMDB dataset (Northcutt et al., 2021), using confidence as defining features and flipped gold labels as target. Specifically, we train a logistic regression model using the scikit-learn implementation (Pedregosa et al., 2011) on the train split of IMDB: we sort the instances by decreasing confidence, sample 300 instances per class and flip the gold labels of the first 150 instances, under the assumption that instances with high confidence are probably correctly labelled.

## 3.3 Datasets

For our data selection experiments, we select 3 commonly used English labelled corpora: IMDB (Maas et al., 2011), SST2 (Socher et al., 2013) and Cola (Warstadt et al., 2019). Additionally, we plot maps for these datasets together with the Yelp (Zhang et al., 2015) and AG News (Gulli, 2005) datasets. We test automatic error detection using two datasets: a version of IMDB which has been re-annotated using both crowdworkers and automatic methods (Northcutt et al., 2021) and the SST2 dataset where we simulated errors by randomly flipping 5% of the gold labels. Table 2 highlights the key facts about these datasets.

## 4 Results

### 4.1 Visual Comparison

We visually compare zero-shot and training dynamics maps, revealing a good similarity between them. As can be seen in Figure 1, while the training dynamics map exhibits more pronounced spikes, reflecting distinct training epochs, the zero-shot maps present a smoother profile. This smoothness stems from the larger number of label description combinations used in the zero-shot setting, in contrast to the few epochs used in the training dynamics. Figure 3 provides a view into the relationship between dataset tasks and the performance of the zero-shot model. For tasks where the zero-shot model yields strong results, such as Yelp Polarity and Ag News, the plotted curve demonstrates a distinct bell shape, which closely mirrors the distribution observed in the training dynamics map. For datasets like Cola, the distribution, although still bell-shaped, shows a larger concentration of instances in the lower half of the graph, indicating a subpar performance of the zero-shot model on this particular task. These observations suggest that the model is highly sensitive to the nature of the task at hand.

### 4.2 Data Selection

For the data selection experiments, we trained models using data samples selected based on the coordinates from both zero-shot and training dynamics maps. The results in Table 3 show that zero-shot maps and training dynamics generally offer comparable performance for data selection, though there are some notable exceptions. For instance, the ambiguous category in the Cola dataset shows a marked difference, which we hypothesize may be due to the inherent complexities of the Cola corpus and limitations in the dual-encoder embedder's ability to assess sentence grammaticality. As such, while zero-shot maps provide a resource-efficient

| DATASET | TASK | CLASSES | TRAIN | TEST | VALIDATION |
|---|---|---|---|---|---|
| Ag news (Gulli, 2005) | topic classification | Business Sci/Tech Sports World | 120,000 | 7,600 | - |
| Cola (Warstadt et al., 2019) | linguistic acceptability | acceptable unacceptable | 8,551 | 1,063 | 1,043 |
| IMDB (Maas et al., 2011) | sentiment | negative positive | 25,000 | 25,000 | - |
| SST2 (Socher et al., 2013) | sentiment | negative positive | 67,349 | 1,821 | 872 |
| Yelp Polarity (Zhang et al., 2015) | sentiment | negative positive | 560,000 | 38,000 | - |

Table 2: Dataset Overview. All the datasets are in English. For the data selection experiments, we use the validation split for Cola and SST2 and the test split for the rest.

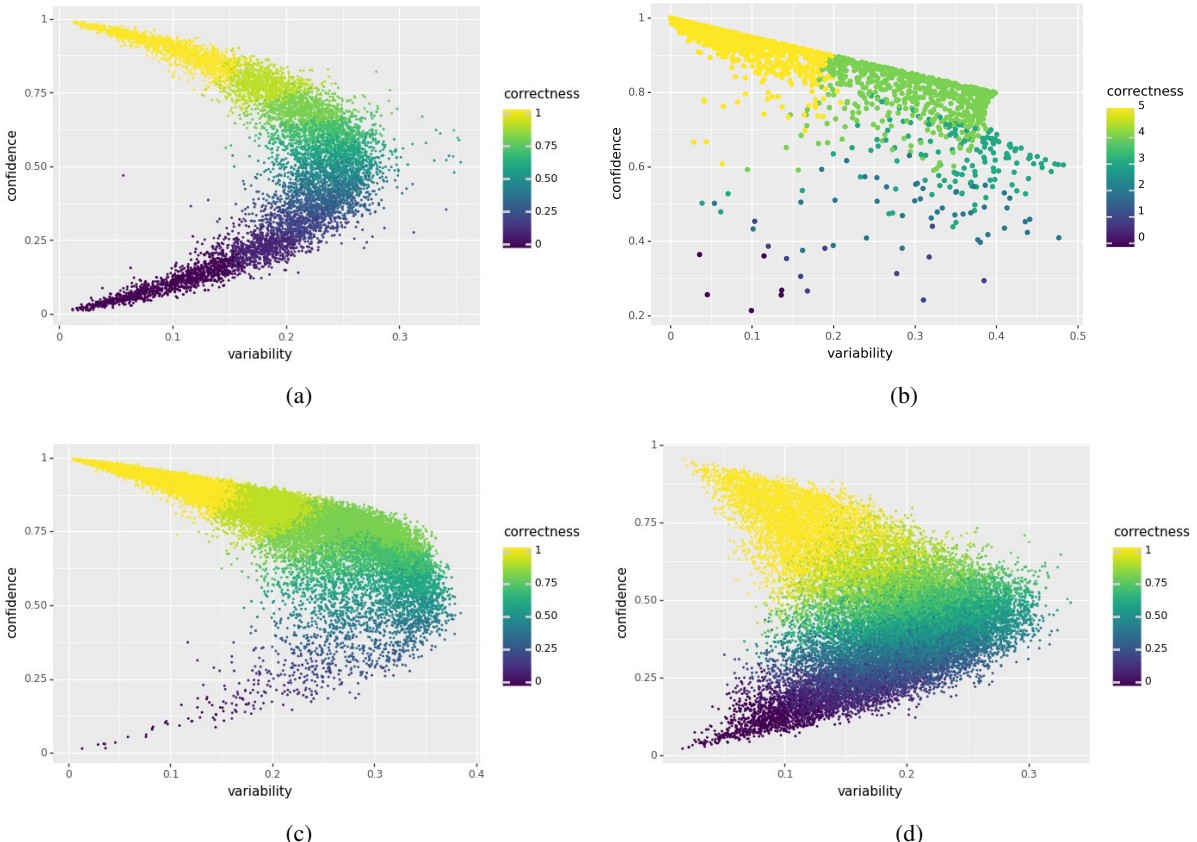

Figure 3: An overview of Zero-shot and Training Dynamics Maps. Top: Cola data maps generated using zero-shot models (left) and training dynamics (right). Bottom: zero-shot maps for the Yelp (left) and Ag News (right) datasets.

| | | COLA | | IMDB | | SST2 | |
|---|---|---|---|---|---|---|---|
| | 100% train | 62.60 | | 92.55 | | 94.04 | |
| | | TD | ZS | TD | ZS | TD | ZS |
| | random | 36.22 | 36.22 | 92.10 | 92.11 | 93.00 | 93.00 |
| | ambiguous | **44.04** | 37.33 | 92.22 | 92.15 | 92.54 | **93.35** |
| | easy-to-learn | 34.13 | 36.41 | **92.24** | 92.25 | 92.43 | 92.43 |
| 33% train | hard-to-learn | 34.43 | 36.01 | 92.23 | 92.24 | 92.43 | 92.43 |
| | high-confidence | 25.12 | 49.18 | 33.38 | 92.22 | 92.89 | 90.02 |
| | high-correctness | 34.42 | **49.43** | 33.77 | 92.21 | 92.89 | 90.37 |
| | low-correctness | 39.40 | 46.09 | 33.53 | 92.14 | **93.23** | 91.63 |
| | low-variability | 34.10 | 36.51 | **92.24** | 92.25 | 92.43 | 92.43 |

Table 3: Macro-averaged F1 scores from Training Dynamics (TD) versus Zero-Shot (ZS) generated maps. For each dataset, the best results for both methods are in bold font.

alternative, they should not be considered a complete substitute for training dynamics, particularly for complex tasks.

We consistently observe significant improvements over the random baseline data selection for both zero-shot and training dynamics methods. Interestingly, for datasets that are annotated by humans, such as Cola and SST2, the advantage over random selection is more pronounced. This may be due to the less systematic labeling of these datasets, as they are annotated by humans, compared to datasets like IMDB which are compiled automatically using proxies like star ratings. These findings suggest that the data selection approach using data

maps is particularly effective when dealing with datasets that exhibit a high degree of labeling complexity.

### 4.3 Error Detection

The automatic error detection results shown are presented in Table 4. Zero-shot maps even outperform Training Dynamics in terms of average precision, showing that this method has potential as a basis for an automatic error detection system.

| MODEL | IMDB | SST2 |
|---|---|---|
| Training Dynamics | 30.2 | 49.0 |
| Zero-Shot | 34.7 | 57.0 |

Table 4: Average Precision scores for the automatic error detection systems on two datasets: the semi-automatically cleaned IMDB dataset and the SST2 dataset where errors were simulated by randomly flipping 5% of the gold labels.

Figure 4a presents a zero-shot map of the IMDB dataset's test split, where instances identified as annotation errors by Northcutt et al. (2021) are highlighted in red. A large proportion of instances with incorrect gold labels clusters towards the lower half of the map. This clustering implies that the confidence of ensemble zero-shot models can be an effective indicator for identifying error instances. Similarly, Figure 4b shows that the average confidence assigned to instances with incorrect annotations is generally lower compared to instances with correct labels.

### 4.4 Computational Requirements

The main advantage of our method is that it bypasses model training. This speedup is further increased by the zero-shot architecture choice, i.e., the bi-encoder network. A comparison of the computation time and resources used for generating training dynamics and zero-shot maps reveals that zero-shot maps are generally much faster to compute, as illustrated in Table 5. The absence of a training phase allows for a different use of resources: given the same GPU, larger — often better — encoding models can be used in the creation of zero-shot maps as there is no requirement for the backward pass typically needed in training.

### 4.5 Discussion

In this study, our goal was to address two key research questions. Firstly, we aimed to create a

|  | TIME | RAM |
|---|---|---|
| Zero-Shot (CPU) | 28 min | 10.6Gb |
| Zero-Shot (GPU) | 1 min | 2.2Gb |
| TD (CPU) | 9 hr 5 min | 16.4Gb |
| TD (GPU) | 14 min | 6.6Gb |

Table 5: Comparison of time and RAM usage for Zero-Shot and Training Dynamics (TD) on both CPU and GPU for a dataset with 10.000 instances using a batch size of 8 and static padding to 512 tokens. RAM is VRAM for the GPU experiments and system RAM for the CPU experiments.

more efficient, lightweight method for constructing data maps (**RQ1**). Our experimental outcomes, as outlined in Sections 4.1 and 4.2, suggest that zero-shot data maps can precisely characterize a dataset when the task can be accurately represented using label descriptions only. Typically, these tasks include most text classification tasks, such as sentiment analysis and topic classification.

Secondly, we focused on automatic error detection (**RQ2**) with an emphasis on adhering to the interpretations of the data maps regions as provided by Swayamdipta et al. (2020), i.e., assuming that hard-to-learn instances are likely to be annotation errors. Our findings in Section 4.3 provide evidence that the confidence scores derived from zero-shot classifiers can indeed serve as a reliable signal for constructing error detection systems. These results suggest that zero-shot maps can be a good alternative to training dynamics maps.

For larger academic datasets, which often require substantial investment in terms of both time and financial resources, the cost associated with training a supervised model could well be justified. In applied settings, particularly in industry, datasets are constantly being created and updated. In these scenarios, we recommend the use of zero-shot data maps as they offer a swift, efficient method for understanding data, while still delivering performance on par with training dynamics maps. With recent advancements in NLP technology, products like ChatGPT (OpenAi, 2022; OpenAI, 2023) have made it much easier to label new data for a wide range of applications. These models have, in some instances, even begun to replace the need for human annotators and crowdsourcing (Gilardi et al., 2023; He et al., 2023). Despite the impressive capabilities of these models in a zero-shot setting, they still commonly make mistakes. Consequently,

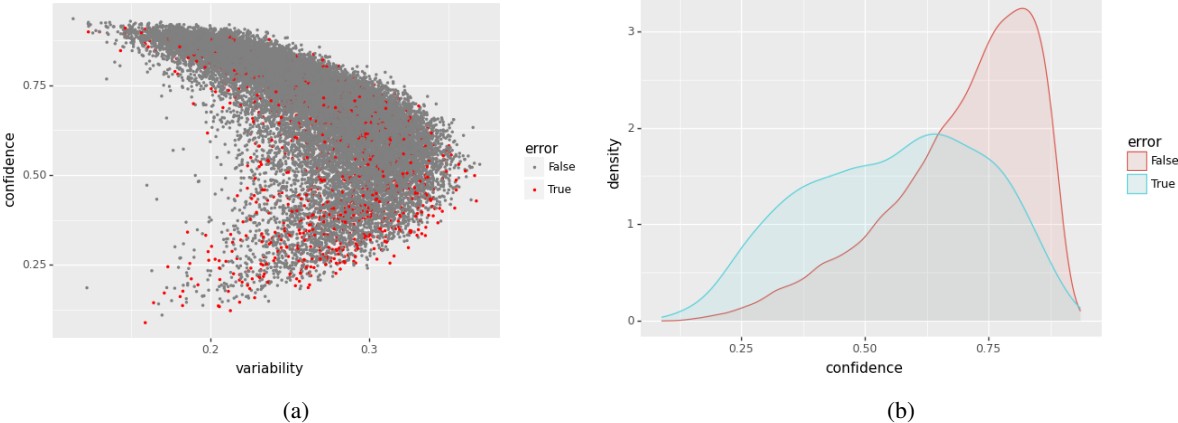

Figure 4: a): A zero-shot data map of the *IMDB* test split. We show in red the annotation errors. b): the confidence distribution for the test split of *IMDB* for instances with a correct gold label (error False) and incorrectly labeled ones (error True)

labeled test sets continue to be a critical requirement for their proper evaluation. This fact underscores the need for fast and accurate data mapping techniques that can be used to identify potential (automatic) annotation errors. This context further highlights the value of the zero-shot data maps proposed in this study.

## 5 Related Work

Recently, several lines of research have presented findings related to the concept of Dataset Cartography. Sanh et al. (2020) demonstrated how weak models, as opposed to strong ones, recreate similar data maps by leveraging annotation artifacts and biases. On the subject of instance difficulty, Ethayarajh et al. (2022) proposed $\mathcal{V}$-*usable information*, a generalization of Shannon information (Shannon, 1948), which provides an aggregate measure of data difficulty in relation to a model; the pointwise $\mathcal{V} - Information$, an extension of $\mathcal{V}$, has been shown to correlate strongly with data map coordinates, and offers the additional benefit of explaining *why* an instance can be challenging for a model. Data maps have been recently applied to a variety of tasks such as syntactic parsing (Kulmizev and Nivre, 2023), hate speech detection (Ramponi and Tonelli, 2022), and visual question answering (Karamcheti et al., 2021). Furthermore, data maps have been shown to be an effective data selection tool for active learning (Zhang and Plank, 2021; Liu et al., 2022). The idea of using semantic similarity for zero-shot classification can be traced back to Gabrilovich and Markovitch (2007). Chang et al. (2008) introduced the concept of dataless classi-

fication by using label descriptions encoded with shallow features for zero- and few-shot classification. Recently, Müller et al. (2022) have shown how modern approaches using dense bi-encoders can outperform more resource-intensive cross-encoder models. For a comprehensive review of automatic annotation error detection, one can refer to Klie et al. (2023).

## 6 Conclusion

In this paper, we have introduced the concept of Zero-Shot Data Maps, a new method for understanding annotated datasets. Our evaluation on various datasets demonstrates the effectiveness of Zero-Shot Data Maps in providing a precise visual representation of labelled datasets.

Our Zero-Shot Data Maps can match the precision of maps produced with training dynamics, while providing the added benefit of being available instantly.

However, training dynamic maps can be drawn for a broader array of tasks, making zero-shot maps as a valuable complement,

rather than a direct replacement. Furthermore, our approach proves to be an effective tool for spotting potential annotation errors: the confidence of ensembled zero-shot models acts as a reliable indicator for identifying incorrect gold labels.

The quality of zero-shot maps largely depends on carefully crafted label descriptions and their ensemble combination. An intriguing direction for future research lies not only in improving the generation of these descriptions, potentially through automated methods, but also in experimenting with

different zero-shot architectures to approach more task types than bi-encoders allow. The combination of our method with active learning and automated error detection algorithms could pave the way for more efficient dataset curation and improvement processes. We believe that our approach to create data maps can help researchers and practitioners improve data quality and ultimately build better models.

## Limitations

The main limitation of our work is that zero-shot data maps from bi-encoders can be compiled effectively only for the same type of tasks that can be solved by bi-encoder network architectures. While other, more flexible but resource-hungry, zero-shot methods can be adapted to build data maps, they cannot compete in terms of speed. Furthermore, the success of zero-shot data maps depends on the quality of label descriptions used in the zero-shot setup.

As a result, zero-shot data maps can also display considerable variability across datasets, and the insights derived from them may not generalize effectively across different tasks.

## Acknowledgments

We thank the anonymous reviewers for their insightful comments and Thomas Mueller for his valuable feedback during the initial stages of this work. The work of Paolo Rosso was in the framework of the FairTransNLP research project (PID2021-124361OB-C31) funded by MCIN/AEI/10.13039/501100011033 and by ERDF, EU A way of making Europe. The work from Symanto has been partially funded by the Pro[2]Haters - Proactive Profiling of Hate Speech Spreaders (CDTi IDI-20210776), the XAI-DisInfodemics: eXplainable AI for disinformation and conspiracy detection during infodemics (MICIN PLEC2021-007681), the OBULEX - *OBservatorio del Uso de Lenguaje sEXista en la red* (IVACE IMINOD/2022/106), and the ANDHI - ANomalous Diffusion of Harmful Information (CPP2021-008994) R&D grants.

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

# A Appendix

## A.1 Model Calibration

The effectiveness of zero-shot maps is contingent on the proper calibration of the zero-shot model. In Figure 5, we demonstrate the impact of calibration, showcasing two maps, one compiled with a calibrated model and the other without. While both maps display a bell-shaped curvature, the curve's prominence is notably enhanced with calibration. As seen from Figure 5, the calibration step is crucial for compiling zero-shot maps that closely resemble those derived using training dynamics.

For the calibration process, we fit an isotonic regressor, which is designed to map our model's initial predicted logits to new, calibrated values. The function used in this process is strictly monotone, which means that it retains the original order of predictions. Consequently, while this calibration process improves the visual representation of our dataset by providing more accurate probability estimates, it does not affect the data selection part of our methodology, since the predicted class labels remain unchanged. Therefore, this calibration process improves the interpretability of our zero-shot data maps without altering their underlying

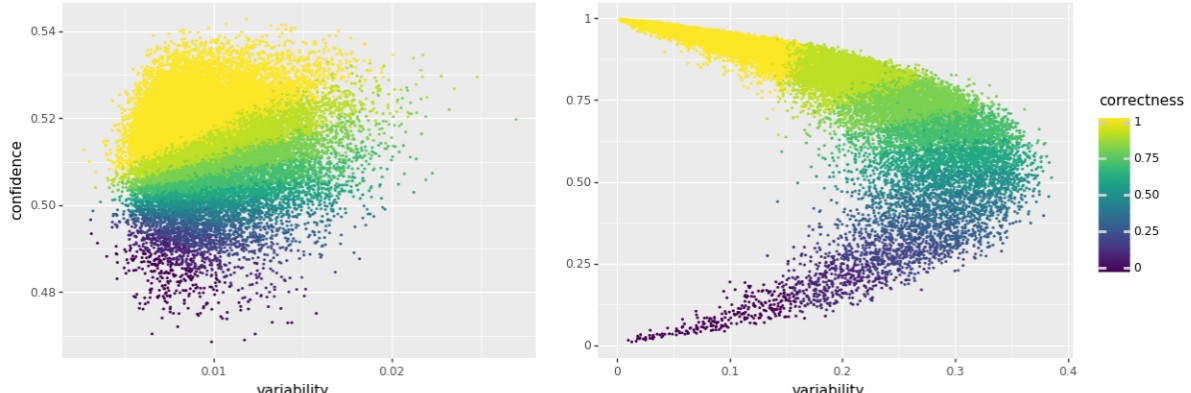

Figure 5: Zero-shot Data Maps for SST2 compiled using *sentence-t5-large* with calibration (right) and without (left).

structure. We note that calibration is done exactly once per encoder and the calibrated model is then applied to all tasks.

## A.2 Analysis of Variability in Label Descriptions

To evaluate the stability of our maps concerning different sets of label descriptions, we introduce *dropout maps*. Starting with the complete coordinate matrix, these maps are created by randomly omitting pairs of label description combinations. We construct dropout maps with varying numbers of retained label description combinations (3, 5, 8, 13, and 21), omitting the rest. A Pearson correlation analysis is then conducted between the variability vectors of the dropout and full maps. Our findings in Table 6 indicate strong robustness, even when only 3 combinations are retained. This confirms that the observed variability in the maps largely stems from label descriptions rather than input instances.

| $n$ | AG_NEWS | COLA | IMDB | SST2 |
|---|---|---|---|---|
| 3 | 0.60 | 0.58 | 0.58 | 0.65 |
| 5 | 0.72 | 0.70 | 0.65 | 0.76 |
| 7 | 0.79 | 0.79 | 0.81 | 0.82 |
| 10 | 0.86 | 0.86 | 0.84 | 0.88 |
| 15 | 0.90 | 0.89 | 0.87 | 0.92 |
| 20 | 0.93 | 0.93 | 0.92 | 0.93 |

Table 6: Pearson correlation coefficients between original full map and dropout maps with varying number of ensemble members.

## A.3 Item-Level Metrics Comparison in Zero-Shot and Training Models

| | AG_NEWS | COLA | IMDB | SST2 |
|---|---|---|---|---|
| confidence | 61.48 | 34.34 | 28.17 | 77.47 |
| correctness | 47.34 | 42.3 | 32.95 | 81.93 |
| variability | 27.82 | 22.8 | 31.88 | 36.34 |

Table 7: Percentage of items categorized into the same bin (high, medium, low) for both training dynamics and zero-shot maps across four datasets. For example, in the SST2 dataset, 77.47% of the items were assigned to the same confidence bin by both methods.

To assess the overlap between training dynamics (TD) and zero-shot (ZS) maps in terms of item categorization, we binned the coordinates into three buckets: high, medium, and low. We then compared the percentage of instances that fell into the same bin in both methods. The results in Table 7 reveal a strong alignment between the TD and ZS approaches. We note that we have conducted here the strictest possible test, measuring the exact matches between the two methods across the whole corpus for each dataset.

## A.4 Implementation Details

We use the Sentence Transformers (Reimers and Gurevych, 2019) and HuggingFace Transformers (Wolf et al., 2020) libraries for using the Transformer models: specifically, for all our experiments we use *sentence-t5-large* (Ni et al., 2022), a 335 million parameter sentence-encoder based on T5 (Raffel et al., 2020). Based on the MTEB benchmark (Muennighoff et al., 2023), *sentence-t5-large* is the best-performing model that we can run on

our GPU and the third best model overall for classification tasks. The AI2-Tango toolkit (Groeneveld et al., 2023) handles our experiment management. We employ the AdamW optimizer in training dynamics experiments and scikit-learn (Pedregosa et al., 2011) for the experiments with frozen embeddings. All the datasets for the experiments are sourced and loaded using the Huggingface Datasets library (Lhoest et al., 2021). All computations are performed on a single RTX A6000 GPU, with a consistent batch size of 8 across all datasets, which is the maximum batch size that can be used to finetune *sentence-t5-large* on our GPU before encountering out-of-memory errors.

## A.5 Dropping Ensemble Members

In the original work involving training dynamics, an epoch burn-out scheme is employed to discard the initial steps until the training process stabilizes: by discarding the predictions from the initial training epochs, this scheme prevents premature influence from the initial, unstable steps of the training.

In our approach to creating zero-shot data maps, we replicate this burn-out effect by dropping ensemble members. This process mirrors the effect of epoch burn-in from the training dynamics method. Specifically, we selectively discard predictions from some ensemble members.

This method ensures that our metrics of confidence and variability are based on more reliable model predictions, thereby enhancing the robustness and reliability of the zero-shot data maps.

## A.6 Label Descriptions

In Table 8, we present an overview of the datasets, labels, corresponding descriptions, and the associated patterns used in our experiments. The label descriptions and patterns were manually crafted, drawing inspiration from previous studies on zero- and few-shot learning (Schick and Schütze, 2021; Müller et al., 2022). . Specifically, we used a template with class names (e.g., 'Category: business, science, sports') to match a realistic scenario where here users would prefer to minimize time spent on crafting descriptions. The label descriptions used in our experiments were generated combinatorially based on the data presented in Table 8.

| Dataset | Label | Descriptions | Patterns |
|---|---|---|---|
| Ag News | Business
Sci/Tech
Sports
World | business, markets, money
science and technology, technology, science
sports, fitness, races
world, global, international | It is {} news., {}, Category: {}., Section {}. |
| Cola | acceptable
unacceptable | acceptable, good, right, perfect
unacceptable, not acceptable, wrong, incorrect | {}, It is {}., This sentence is grammatically {}.
This sentence is {}., The grammaticality of this sentence is {}. |
| IMDB | neg
pos | negative, bad, terrible, poor
positive, great, awesome, fantastic | {}, This feels {}., Just {}!, All in all, the movie was {}. |
| SST2 | negative
positive | negative, bad, terrible, poor
positive, great, awesome, fantastic | {}, This feels {}., Just {}!, All in all, the movie was {}. |
| yelp polarity | 1
2 | negative, bad, terrible, poor
positive, great, awesome, fantastic | {}, This feels {}., Just {}!, All in all, the restaurant was {}. |

Table 8: Overview of the labels, label descriptions and patterns used for each dataset. The placeholder {} in the patterns host the label descriptions. In the Label column we include the class names as they are encoded in the datasets accessed through the Huggingface Datasets (Lhoest et al., 2021) library.