# OpenReview forum: "Zero-Shot Data Maps. Efficient Dataset Cartography Without Model Training"
_EMNLP/2023/Conference — EMNLP 2023 Findings_

### Official Review · Reviewer_b4Jj · 2023-08-02

**Soundness:** 5

**Excitement:**

4: Strong: This paper deepens the understanding of some phenomenon or lowers the barriers to an existing research direction.

**Paper Topic And Main Contributions:**

The paper introduces Zero-Shot Data Maps, a novel approach for diagnosing large annotated datasets without any model training. The key idea is to replace model training, which is time-consuming, with a strong and efficient zero-shot model. Like Data maps, the confidence and variability over the true labels are computed. The paper then presents a comparative analysis between training dynamics and the proposed approach on various datasets. The findings show that Zero-Shot Data Maps match Data Maps while achieving up to a 14x speedup.

**Reasons To Accept:**

1. Given the recent trend of zero/few-shot methods in NLP, the problem of interest is interesting and timely.
2. The method is well-motivated and simple, and the results are impressive. This efficiency enhancement makes the technique more practical and scalable for large datasets.
3. Evaluation is comprehensive, covering visual comparison and downstream applications (e.g., data selection and error detection).

**Reasons To Reject:**

1. It would be more interesting to see if LLMs can be applied in this setting and if they make the story different, even on a small scale.

**Reproducibility:**

4: Could mostly reproduce the results, but there may be some variation because of sample variance or minor variations in their interpretation of the protocol or method.

**Reviewer Confidence:**

4: Quite sure. I tried to check the important points carefully. It's unlikely, though conceivable, that I missed something that should affect my ratings.

---

> ### Author Rebuttal · Authors · 2023-08-29
>
> Thanks for your review.
>
> While we like the idea of adapting our methods to work with LLMs and have played with it, the final system would considerably compromise the efficiency we aim for with our current method: to get N different responses on a single data point, we would need to prompt the LLM N times with N different prompts.

---

### Official Review · Reviewer_t41N · 2023-08-03

**Typos Grammar Style And Presentation Improvements:** 1. Figure 2 is cited before Figure 1 …
**Soundness:** 2

**Excitement:**

2: Mediocre: This paper makes marginal contributions (vs non-contemporaneous work), so I would rather not see it in the conference.

**Paper Topic And Main Contributions:**

This paper proposes a method for building zero-shot data maps. Data Maps is useful for diagnosing large annotated datasets. It visualize the data points according to the confidence (the average probability assigned by the ensemble members to a specitfic label) and variability / uncertainty (the standard deviation of the model’s probabilities across the ensemble members) scores. In previous work (Swayamdipta et al., 2020), confidence and variability are calculated using training dynamics, which requires the fitting of a strong model to the dataset. This paper proposes to compute confidence on the true label and variability over the members of an ensemble of zero-shot models constructed with different — but semantically equivalent — label descriptions, i.e., textual representations of each class in a given label space, based on sentence-t5-large. Emprical results show that the zero-shot data maps (after calibration) generally match those produced by the traditional method while delivering up to a 14x speedup.

**Questions For The Authors:**

1. How is the isotonic regressor trained for caliration? Does this procedure require some sort of ground-truth labels?

2. How is the final data map sensitive to the label description phrasing? Section 2.2 does not dispel my concerns.

**Reasons To Accept:**

1. The framework proposed in this paper is relatively simple.

2. The method seems to be practical, at least based on the reported results.

**Reasons To Reject:**

1. The method proposed by Swayamdipta et al. (2020) is based on training dynamics, which provides a certain level of theoretical interpretability (variability captures the model's uncertainty). While the zero-shot data map involves too many subjective factors, like the design of label description phrasing and the calibration. This may potentially have a negative impact on the confidence of applying this method in practical applications.

2. The authors claim that the calibration function is strictly monotone and thus does not alter their underlying structure in Appendix A.1. However, the effect of calibration may have a significant impact when we need quantitative analysis or comparison between different categories.

**Reproducibility:**

3: Could reproduce the results with some difficulty. The settings of parameters are underspecified or subjectively determined; the training/evaluation data are not widely available.

**Reviewer Confidence:**

3: Pretty sure, but there's a chance I missed something. Although I have a good feel for this area in general, I did not carefully check the paper's details, e.g., the math, experimental design, or novelty.

---

> ### Author Rebuttal · Authors · 2023-08-29
>
> Thanks for your review.
>
> **On Calibration.**
>
> We want to emphasize that calibration is performed once for each encoder, not for every dataset/task. Specifically, we calibrate the zero-shot classifier using sentence-t5-large and apply it across all experiments. In practice, our goal is to provide users with a pre-calibrated zero-shot classifier: we believe that this configuration adds no subjective factors or negative effects on confidence.
>
> Here is the calibration process in details:
>
> - Initial Predictions. We start by obtaining predictions from the uncalibrated zero-shot classifier on a collection of labeled datasets.
> - Optimal Probability Estimation. For every instance, an optimal probability is estimated. Instances are organized based on their initial softmax probability. In this way, a neighborhood of 100 instances surrounding the current one is identified. The optimal probability for an instance is then calculated using the accuracy of this neighborhood, which relies on the ground-truth labels.
> - Logits Binning. All prediction logits are split uniformly to create bins. Logits are assigned to these bins using binary search.
> - Function Formulation and Weight Estimation. The calibration adjusts the initial logits using a monotonic function: $y=f(x)=x \times wi(x)$​. Here, $x$ represents the initial logit and $y$ is the updated logit. The function $i(x)$ assigns a specific bin to $x$, while $wi​$ is a weight for bin $i$. These weights, also termed as scale factors (SF), are estimated to minimize the difference between the ideal probability and the softmax probability post $f(x)$ application.
>
> Will will upgrade the appendix to include all these details.
>
> **On Map sensitivity to label description phrasing.**
> I believe that this question is the same question asked by reviewer  CCvW. I report here the results from the experiments we conducted to address this point.
>
> We have conducted new experiments to better understand the effect of label description variability on our map's structure.
> - Data Processing: For each dataset we have been working with, the coordinate matrix was derived. This matrix offers a 2D representation, the columns of which indicate 'confidence', 'variability', and 'correctness' and the rows are individual instances. This is the data map that we use for plotting the images in the paper.
> - Introduction of Dropout Maps: 1000 dropout maps were constructed by randomly keeping various numbers of combinations of label descriptions (3, 5, 8, 13, and 21): e.g., in the table below, n=3 indicates that we build a new map with only 3 combinations of label descriptions. We note that for some datasets we have more than 700 combinations.
> - Correlation Assessment: We then ran a correlation analysis (Pearson) with the original variability vector from the full map against the variability vector from every dropout map.
> Our analysis reveals a strong correlation between the original full map and these dropout maps, as will be detailed in the table below. This correlation underscores the robustness of our approach even in the face of potential variability due to label descriptions. We note that already in the extreme case where we build a map with only 3 combinations, there is a good to strong correlation with the full map.
>
> |   n |   ag_news |   cola |   imdb |   sst2 |
> |----:|----------:|-------:|-------:|-------:|
> |   3 |      0.6  |   0.58 |   0.58 |   0.65 |
> |   5 |      0.72 |   0.7  |   0.65 |   0.76 |
> |   7 |      0.79 |   0.79 |   0.81 |   0.82 |
> |  10 |      0.86 |   0.86 |   0.84 |   0.88 |
> |  15 |      0.9  |   0.89 |   0.87 |   0.92 |
> |  20 |      0.93 |   0.93 |   0.92 |   0.93 |

---

### Official Review · Reviewer_FBi8 · 2023-08-09

**Soundness:** 2

**Excitement:**

2: Mediocre: This paper makes marginal contributions (vs non-contemporaneous work), so I would rather not see it in the conference.

**Paper Topic And Main Contributions:**

 Authors introduce a novel method for compiling data maps up to 14x faster. They compare their method against training dynamics-based maps and show that zero-shot maps can be used for automatic annotation error detection.

**Questions For The Authors:**

1. I am curious about using larger and stronger classifier.
2. Could you provide a more detailed error analysis?

**Reasons To Accept:**

The paper is easy to follow.
Proposed method is simple but effective.

**Reasons To Reject:**

Overall, the model proposed by the author is very simple and straight forward, which might limit the technical contributions of the model.

**Reproducibility:**

4: Could mostly reproduce the results, but there may be some variation because of sample variance or minor variations in their interpretation of the protocol or method.

**Reviewer Confidence:**

3: Pretty sure, but there's a chance I missed something. Although I have a good feel for this area in general, I did not carefully check the paper's details, e.g., the math, experimental design, or novelty.

---

> ### Author Rebuttal · Authors · 2023-08-29
>
> Thanks for your review.
>
> Regarding a larger and stronger classifier: to build our classifier we use sentence-t5-large, which is already a SOTA encoder model in its class. We have also conducted several experiments with weaker encoders and, as long as the model is calibrated, results are in line with what we would expect to see.
>
> Regarding simplicity, one of the main goals of the paper was indeed from the beginning to reproduce data maps with a simpler system. I would argue that it is a strong point of the paper, agreeing with the reviewer, who also mentioned it as a reason for accepting.

---

### Official Review · Reviewer_CCvW · 2023-08-14

**Soundness:** 4

**Excitement:**

4: Strong: This paper deepens the understanding of some phenomenon or lowers the barriers to an existing research direction.

**Paper Topic And Main Contributions:**

This paper introduces Zero-Shot Data Maps, a  method for visualizing and interpreting labeled datasets without the need for model training. These 2D Data Maps [1] helps in interpreting dataset, by finding different types of instances(such as easy-to-learn or annotation errors). The method utilizes a single bi-encoder model with varying label descriptions to estimate the confidence and variability for each data point. The authors demonstrate that these Zero-Shot Data Maps can approximate Training Dynamics maps, particularly in tasks such as error detection, while being significantly faster to compute. In summary, the key contributions of this work are the introduction of an efficient zero-shot approach to data cartography, validation of its effectiveness as an alternative to training-based maps.

**References**
1. Swayamdipta, S., Schwartz, R., Lourie, N., Wang, Y., Hajishirzi, H., Smith, N.A., & Choi, Y. (2020). Dataset Cartography: Mapping and Diagnosing Datasets with Training Dynamics. Conference on Empirical Methods in Natural Language Processing.

**Questions For The Authors:**

In addition to the points raised in the "Reasons to Reject" section, I would like the authors to address the following queries:

D. **Interpretation of Table 3:** In lines 383-385, you state: "The results in Table 3 show that both methods yield comparable outcomes when used for data selection." However, for some categories, the difference between the two columns appears quite substantial. For instance, consider the 'high confidence' category for the CoLA dataset. Could you please elaborate on this?

E. **Comparison of Zero-Shot Maps to Training Dynamics:** A detailed comparison between individual instances in Zero-Shot (ZS) and Training Dynamics (TD) maps would have been insightful. For example, it would be interesting to know what percentage of instances labeled as 'high-confidence' in ZS maps also fall under the same category in TD maps. Have you conducted some similar study? If so, incorporating these findings into the paper would be beneficial.

F. **Computational Complexity with Label Descriptions:** In lines 216-217, you claim that the proposed method "delivers efficient inference time regardless of the number of labels". Is this efficiency due to the fact that label descriptions need to be encoded only once for the entire dataset? However, depending on the number of label descriptions required per class, encoding these descriptions might also be computationally expensive. Further few-shot datasets may have small ratio of input-instances/number of labels. Could you include a more detailed analysis of the computational complexity, considering both the number of labels and label descriptions as variables?

**Reasons To Accept:**

The paper presents a novel concept of Zero-Shot Data Maps, offering a more efficient approach to analyzing large datasets compared to traditional methods that rely on training dynamics. This contribution is significant, as it enables more expedient evaluation of extensive datasets. The authors substantiate the effectiveness of Zero-Shot Data Maps through qualitative analysis and downstream tasks, demonstrating their similarity to maps generated through training dynamics. Consequently, these Zero-Shot Data Maps can serve as a viable alternative for tasks such as error detection.

**Reasons To Reject:**

A. **Label Descriptions:** The paper lacks some clarity on the process of creating label descriptions used in the zero-shot setup. Specifically, it does not detail the strategies employed to craft these descriptions or assess their quality. Does Table 6 contains all the descriptions? A comprehensive explanation of these aspects would enhance the understanding of the method's effectiveness and its impact on the resulting data maps.

B. **Variability in Label Descriptions:** The paper does not sufficiently address the potential impact of variability in label descriptions on the generated data maps. Specifically, it is unclear how the authors ensure that the observed variability in data maps is attributed to the descriptions themselves rather than the input instances. For instance, high variability could potentially be a result of poorly crafted descriptions. An analysis to disentangle these effects would strengthen the paper's claims and ensure its generalizability to larger datasets.

C. **Error Detection:** The evaluation of the automatic error detection capability of the proposed method is limited to a single dataset (IMDB). Given the significance of demonstrating that the error detection performance is on par with methods using Training Dynamics, it would be beneficial to include evaluations on additional datasets. This would provide a more robust validation of the method's effectiveness in error detection.

**Reproducibility:**

5: Could easily reproduce the results.

**Reviewer Confidence:**

3: Pretty sure, but there's a chance I missed something. Although I have a good feel for this area in general, I did not carefully check the paper's details, e.g., the math, experimental design, or novelty.

---

> ### Author Rebuttal · Authors · 2023-08-29
>
> Thanks for your review.
>
> **A. Label Descriptions**
>
> In appendix A.4, we outline our method for creating label descriptions based on previous work. We used a template with class names (e.g., 'Category: {business, science, sports}') because it seems practical, considering users might want to spend less time on creating descriptions. All the label descriptions we used in our experiments are generated combinatorially from the data listed in Table 6.
> In Section 2.2 of our paper, we mentioned that even if a label description seems well crafted, it might not always give the best results. We have shown the performance of individual label descriptions in Table 1 and Figure 2.
> To test our approach further, we tried a different way of creating label descriptions. We used ChatGPT (3-5 Turbo, consulted on August 26, 2023) to generate new descriptions for the CoLA dataset. After manually checking and testing these descriptions, the results were similar to our original ones. We plan to add this data in our paper's next version.
>
> **B. Variability in Label Descriptions**
>
> To address this point, we have conducted new experiments to better understand the effect of label description variability on our map's structure.
>
> - Data Processing: For each dataset we have been working with, the coordinate matrix was derived. This matrix offers a 2D representation, the columns of which indicate 'confidence', 'variability', and 'correctness' and the rows are individual instances. This is the data map that we use for plotting the images in the paper.
> - Introduction of Dropout Maps: 1000 dropout maps were constructed by randomly keeping various numbers of combinations of label descriptions (3, 5, 8, 13, and 21): e.g., in the table below, n=3 indicates that we build a new map with only 3 combinations of label descriptions. We note that for some datasets we have more than 700 combinations.
> - Correlation Assessment: We then ran a correlation analysis (Pearson) with the original variability vector from the full map against the variability vector from every dropout map.
> Our analysis reveals a strong correlation between the original full map and these dropout maps, as will be detailed in the table below. This correlation underscores the robustness of our approach even in the face of potential variability due to label descriptions. We note that already in the extreme case where we build a map with only 3 combinations, there is a good to strong correlation with the full map.
>
> |   n |   ag_news |   cola |   imdb |   sst2 |
> |----:|----------:|-------:|-------:|-------:|
> |   3 |      0.6  |   0.58 |   0.58 |   0.65 |
> |   5 |      0.72 |   0.7  |   0.65 |   0.76 |
> |   7 |      0.79 |   0.79 |   0.81 |   0.82 |
> |  10 |      0.86 |   0.86 |   0.84 |   0.88 |
> |  15 |      0.9  |   0.89 |   0.87 |   0.92 |
> |  20 |      0.93 |   0.93 |   0.92 |   0.93 |
>
>
> **C. Error Detection**
>
> Given that large annotated corpora are usually not re-annotated or checked for errors independently, they are hard to find. We therefore report here an additional experiment using synthetic data. We randomly perturb 5% of the labels of the sst2 dataset and report the average precision in addition to the existing results obtained on the semi-automatically reviewed version of IMDB. Zero-shot maps outperform training dynamic maps by a large margin.
>
>
> |                  | imdb | sst2 |
> |------------------|------|------|
> | training dynamics| 30.2 | 49.0 |
> | zero shot        | 34.7 | 57.0 |
>
>
> **D. Interpretation of Table 3**
>
> We hypothesize that the differences in performance, especially in the high-confidence category, might be rooted in the inherent challenges of the CoLA corpus. Specifically, a dual-encoder embedder might not possess adequate knowledge to accurately understand sentence grammaticality: this hypothesis is supported by the low performance of the classifier in this dataset.This potential limitation of the dual-encoder could be contributing to the observed discrepancies between the two methods for the CoLA dataset.
> We will rewrite the sentence and make sure to frame this hypothesis appropriately in our paper for clearer interpretation. As we mentioned in the limitations section, not all tasks can be properly modeled by this framework and zero-shot maps are an efficient complement to training dynamics maps, more than a replacement.
>
>
> **E. Comparison of Zero-Shot Maps to Training Dynamics**
>
> To address this question we have binned the coordinates in 3 bins (intuitively: high, low, medium) and compared the percentage of instances that fall in the same bin. We observe good to very strong agreement between the two methods. We note that we have conducted here the strictest possible test, measuring the exact matches between the two methods across the whole dataset for each corpus.
>
> | feature     |   ag_news |   cola |   imdb |   sst2 |
> |:------------|----------:|-------:|-------:|-------:|
> | confidence  |     61.48 |  34.34 |  28.17 |  77.47 |
> | correctness |     47.34 |  42.3  |  32.95 |  81.93 |
> | variability |     27.82 |  22.8  |  31.88 |  36.34 |
>
>
>
>
>
> **F. Computational Complexity with Label Descriptions**
>
> Zero-shot dual-encoders are efficient at inference time because a single pass is required over the input data and the label descriptions, independently of the number of labels. Given that data maps are most useful for analyzing large annotated datasets, the number of instances is always going to exceed the number of labels; furthermore, label descriptions are usually much shorter than input texts. However, I have computed the required time (seconds) to encode different datasets using Colab (Nvidia T4 GPU) with sentence-t5-large. We have simulated a worst case scenario: for both texts and labels we have sampled random texts from the IMDB corpus, meaning that label descriptions are here also quite long.
>
> |   n_instances/n_labels |     3 |     5 |     7 |    13 |    50 |   100 |   1000 |
> |--------------:|------:|------:|------:|------:|------:|------:|-------:|
> |             3 |  0.24 |  0.31 |  0.37 |  0.59 |  1.85 |  3.45 |  32.01 |
> |             5 |  0.32 |  0.37 |  0.44 |  0.61 |  1.93 |  3.53 |  32.01 |
> |             7 |  0.37 |  0.45 |  0.5  |  0.71 |  1.98 |  3.62 |  32.22 |
> |            13 |  0.59 |  0.63 |  0.71 |  0.9  |  2.16 |  3.79 |  32.19 |
> |            50 |  1.7  |  1.76 |  1.87 |  2.18 |  3.32 |  4.91 |  33.57 |
> |           100 |  3.24 |  3.36 |  3.47 |  3.62 |  4.75 |  6.41 |  34.96 |
> |          1000 | 32.04 | 32.21 | 32.5  | 32.63 | 33.81 | 35.46 |  69.99 |

---

### Meta-Review · Area_Chair_kdsT · 2023-09-15

**Recommendation:** 3

**Metareview:**

This paper presents Zero-Shot Data Maps based on a single bi-encoder model with diverse label descriptions to assess the confidence and variability of each data point. The authors illustrate that Training Dynamics maps can be approximated using  Zero-Shot Data Maps while offering significantly faster computation.


Reviewers are divided on this submission. I think the review by Reviewer FBi8 has a low quality and needs to be discarded. While the remaining two reviews have acceptable qualities, the two reviews with positive opinions are more extensive and the mentioned strengths outweigh the weaknesses raised by the negative review. I have read the reviews and the follow-up discussions and I think it is reasonable to accept this paper into the Findings track.

---

### Decision · Program_Chairs · 2023-10-07

**Decision:**

Accept-Findings

**Comment:**

This paper presents Zero-Shot Data Maps based on a single bi-encoder model with diverse label descriptions to assess the confidence and variability of each data point. The authors illustrate that Training Dynamics maps can be approximated using  Zero-Shot Data Maps while offering significantly faster computation.


Reviewers are divided on this submission. I think the review by Reviewer FBi8 has a low quality and needs to be discarded. While the remaining two reviews have acceptable qualities, the two reviews with positive opinions are more extensive and the mentioned strengths outweigh the weaknesses raised by the negative review. I have read the reviews and the follow-up discussions and I think it is reasonable to accept this paper into the Findings track.